# Leucine-Rich Repeat Kinase 1 Signaling Targets Proteins Critical for Endosome/Lysosome Sorting and Trafficking in Osteoclasts [note 1]

**DOI:** 10.3390/biology14040326

**Published:** 2025-03-24

**Authors:** Weirong Xing, Yian Chen, Anakha Udayakumar, Haibo Zhao, Subburaman Mohan

**Affiliations:** 1The Musculoskeletal Disease Center, Jerry L Pettis VA Medical Center, Loma Linda, CA 92357, USA; yian.chen2@va.gov (Y.C.); anakha.udayakumar@va.gov (A.U.); subburaman.mohan@va.gov (S.M.); 2Department of Medicine, Loma Linda University, Loma Linda, CA 92354, USA; 3Graduate Schools, Loma Linda University, Loma Linda, CA 92354, USA; 4Southern California Institute for Research and Education, Long Beach, CA 90815, USA; haibo.zhao@va.gov

**Keywords:** *Lrrk1*, knockout mice, osteopetrosis, proteomics, phosphoprotein, cofilin 1, lysosome, endosome sorting, osteoclast, VPS35

## Abstract

Two types of cells located in the bone regulate bone mass. Osteoclasts cause bone destruction whereas osteoblasts produce new bone matrix. Bone loss occurs with age partially because the rate of bone formation cannot compensate for the increase in bone degradation because of sex hormone deficiency and other causes. Our previous studies demonstrated that LRRK1 played a critical role in regulating osteoclast-mediated bone resorption. Mice lacking LRRK1 manifested high bone mass because of the failure of osteoclasts to resorb bone. The molecular mechanism of LRRK1 regulation of osteoclast function is elusive. To identify potential LRRK1 targets in osteoclasts, we performed a 2D phosphor-proteomics analysis with lysates extracted from osteoclasts derived from *Lrrk1* gene knockout and wild-type mice. Differentially phosphorylated peptide spots between the two types of cells were collected, digested with trypsin, and identified by mass spectrometry. A total of 6 out of 17 differentially expressed phosphoproteins are known to be implicated in endosome/lysosome sorting, vacuolar protection, and trafficking. SNX2, VPS35, VTA1, CFL1, and CTSA were highly hypophosphorylated whereas SNX3 was hyperphosphorylated in LRRK1-deficient osteoclasts. Further Phos-tag SDS PAGE analyses confirmed the downregulation of VPS35 and CFL1 phosphorylation in LRRK1-depleted osteoclasts. Our findings suggest that LRRK1 signaling regulates osteoclast function via phosphorylating VPS35 and CFL1 and modulating endosome/lysosome distribution and F-actin remodeling in osteoclasts.

## 1. Introduction

Osteoporosis is a skeletal disease that occurs with aging because of reduced bone mineral density (BMD) and/or altered bone structure or quality because of high bone turnover states in which the rate of new bone formation cannot compensate for the rate of osteoclast-mediated bone loss. The disease can reduce bone strength, deteriorate bone microarchitecture, and cause bones to become brittle, leading to increased susceptibility to fracture [1]. Therefore, early intervention of age-related bone loss and prevention of osteoporotic fractures by suppressing osteoclast bone resorptive function and/or enhancing bone formation are needed to reduce the healthcare burden and improve patients’ quality of life. Currently, most available antiresorptive medications, such as bisphosphonates and monoclonal anti-RANKL antibodies, slow down the rate of osteoclast-mediated bone turnover. However, treatments of patients with these medications also lead to inhibition of osteoblast-mediated bone formation. Long-term antiresorptive therapy with bisphosphonates may cause atypical femur fractures, impede tooth movement, increase serious osteonecrosis risks in the alveolar bones of the mandible and the maxillae, and compromise fracture healing [2,3,4,5,6]. Given the current limitation of the anabolic drugs and the side effects of the anti-resorptive agents, there is a critical need for the identification of new drug targets and the development of novel alternative medications that suppress osteoclast activity but do not compromise osteoclast differentiation and bone formation.

Among over 4500 gene knockout (KO) mouse lines, several lines were identified to have high bone density [7]. Of these, leucine-rich repeat kinase 1 (LRRK1) KO mice exhibited the highest BMD [8]. Mice without the *Lrrk1* gene present with severe osteopetrosis, a disease that causes bones to become overly dense and is the opposite of osteoporosis in vertebral and tubular bones because of the functional failure of osteoclasts to resorb bone [8]. We are particularly interested in LRRK1 because LRRK1 is an ideal drug target, and partial inhibition of the LRRK1 function with small molecular inhibitors is assumed to enhance bone density and strength for the treatment of osteoporosis. In our previous studies, we demonstrated that osteoclasts lacking LRRK1 were defective in RANKL-induced cytoskeletal rearrangement, had weak peripheral sealing zones on bone, and could not degrade the bone matrix. *Lrrk1* KO mice responded normally to anabolic intervention with parathyroid hormone-related protein and resisted estrogen deficiency-induced bone loss [8]. In our previous studies on the mechanism of LRRK1 action in osteoclasts, we found that LRRK1-deficient osteoclasts, unlike the wild-type (WT) osteoclasts, had altered trafficking of the acidic vacuoles/lysosomes [9]. The lysosomes were distributed in the cytoplasmic compartment away from the extracellular lacunae of the bone. Although cathepsin K and v-ATPase expression were comparable in LRRK1-deficient osteoclasts, these lysosomal-associated proteins were not localized on the ruffled border and were not secreted. These data provide evidence that LRRK1 positively directs osteoclast function via modulating lysosomal trafficking and protease/acid exocytosis. Since LRRK1 is a serine/threonine kinase and most aspects of cellular biology and protein function are regulated by protein phosphorylation, we hypothesized that LRRK1 regulated lysosome distribution and osteoclast function via directly or indirectly phosphorylating proteins that are critical for lysosomal movement. To test our hypothesis, we performed a 2D DIGE phosphor-proteomics analysis to identify potential LRRK1 targets in osteoclasts.

## 2. Materials and Methods

### 2.1. Cell Line, Plasmid, Recombinant Proteins, and Antibodies

CMG14-10 cells producing recombinant macrophage colony-stimulating factor (M-CSF) was kindly provided by Dr. Sunao Takeshita and reported in [10]. Plasmid pGEX-4T-1 mRANKL 158-316 was a gift from Dr. Steven Teitelbaum at Washington University School of Medicine (St. Louis, MO, USA) [11]. The RANKL-GST fusion was produced in *E. coli* and purified by Bio-Scale^TM^ ProfinityM GST cartridge and HiQ Biorad ion exchange column by Next Generation of Chromatography (Bio-Rad, Hercules, CA, USA). Monoclonal antibody specific to β-actin (P3555) was obtained from Sigma (St. Louis, MO, USA). Polyclonal antibodies to Actin Reorganization and Phospho-Akt Pathway Antibody Sampler Kits (#9967S and #9916S) including pS3-Cofilin (77G2), pan-Cofilin (D3F9) XP, pan-VASP, pS157-VASP, p239-VASP, pS380-PTEN, pT308-AKT (D25E6), pS473-AKT (D9E), pan-AKT (C67E7), and pS9-GSK3β (D85E12) rabbit antibodies were purchased from Cell Singnaling Technology, Inc. (Danvers, MA, USA). Monoclonal anti-VTA1 (D6) and anti-SNX2 (F-8) antibodies (sc-374012 and sc-390510) were from Santa Cruz Biotechnology, Inc. (Santa Cruz, CA, USA). Polyclonal anti-VPS35 (#10236-1-AP) antibody was from Proteintech (Rosemont, IL, USA). Primary antibodies from Cell signaling Technology were immunoblotted at a dilution of 1:1000, and others were diluted at 1:500 as recommended by the manufacturers. The HRP-conjugated secondary antibodies (A9169-2ML and A5278-1ML) against the primary rabbit polyclonal and mouse monoclonal antibodies, respectively, were purchased from Sigma and were diluted at 1:15,000 for immunoblotting.

### 2.2. Mice, Primary Osteoclast Cultures, and Protein Extraction

*Lrrk1* KO and control WT mice were generated as described [8]. Mice were housed at the VA Loma Linda Healthcare System (VALLHCS) under standard approved laboratory conditions with controlled illumination (14 h light, 10 h dark), temperature (22 °C), and unrestricted food and water. Animal experiments were performed with the approval of the Institutional Animal Care and Use Committee of VALLHCS (#IACUC, Xing 0005/1442). Osteoclast precursors derived from the spleens of 4-week-old male (*n* = 3), female (*n* = 3) *Lrrk1* KO, and corresponding age- and gender-matched WT littermate mice were differentiated in αMEM medium supplemented with 10% FBS, 100 µg/mL streptomycin, 100 units/mL penicillin, 2% CMG14-12 cell conditioned medium containing recombinant M-CSF at 1 µg/mL (e.g., at 20 ng/mL final concentration), and 200 ng/mL RANKL-GST fusion protein for 5 days [10]. To prevent mature osteoclasts from undergoing apoptosis, we only starved the cells in M-CSF and RANKL-free media for 1 h, followed by 1 h of M-CSF and RANKL restimulation. Cells were lysed in a lysis buffer containing 20 mM Bicine buffer (pH 7.5), 0.6% CHAPS, 1 mM DTT, 1% protease inhibitors cocktail, and 1% phosphatase inhibitors cocktail (Sigma) on ice for 20 min, and the total cellular protein was collected after centrifuging at 12,000 rpm for 10 min.

### 2.3. Comparative Analysis of Phosphoprotein Expression

Two-dimensional difference gel electrophoresis (2D DIGE) phosphor-proteomics analysis and protein identification were performed by Applied Biomics, Inc. (Hayward, CA, USA). Briefly, a protein sample buffer was exchanged for a 2D lysis buffer (30 mM Tris-HCl, pH 8.8, containing 7 M urea, 2 M thiourea, and 4% CHAPS). The extracted protein from *Lrrk1* KO and WT osteoclasts was combined, and the protein concentration was measured using a Bio-Rad Protein Assay Kit II (#500-0002) according to the manufacturer’s instructions (Bio-Rad). For the phospho profiling, 250 µg of cellular protein from *Lrrk1* KO or WT osteoclasts was mixed with 1.0 µL of diluted Cy3-dye and kept in the dark on ice for 30 min. The labeling reaction was stopped by adding 1.0 µL of 10 mM lysine to each sample and incubating it in the dark on ice for an additional 15 min. The labeled protein was run on a 2D SDS PAGE system. The gels were then stained using Pro-Q^®^ Diamond Phosphoprotein Gel Stain according to the manufacturer’s protocol (Life Technologies Biotech Company, Carlsbad, CA, USA), followed by scanning using a Typhoon TRIO system, and finally analyzed using DeCyder software version 6.0 (GE Healthcare, Chicago, IL, USA).

For the protein profiling, 30 µg cellular protein from *Lrrk1* KO and control WT cells was labeled with Cy3- and Cy5-dye, respectively. The Cy3-labeled *Lrrk1* KO and Cy5-labeled WT proteins were mixed. The 2X 2D sample buffer (8 M urea, 4% CHAPS, 20 mg/mL DTT, 2% pharmalytes and a trace amount of bromophenol blue), 100 µL Destreak solution and rehydration buffer (7 M urea, 2 M thiourea, 4% CHAPS, 20 mg/mL DTT, 1% pharmalytes, and a trace amount of bromophenol blue) were added to the labeling mix to make a total volume of 250 µL. The mixture of two labeled proteins was loaded into the same IEF strip holder and run on the same gel (pH 3–10 linear) according to Amersham BioSciences’ instructions (Amersham BioSciences Corp., Piscataway, NJ, USA). Upon finishing the IEF, the IPG strips were incubated in the freshly made equilibration buffer-1 (50 mM Tris-HCl, pH 8.8, containing 6 M urea, 30% glycerol, 2% SDS, a trace amount of bromophenol blue and 10 mg/mL DTT) for 15 min with gentle shaking. The strips were then rinsed in freshly made equilibration buffer-2 (50 mM Tris-HCl, pH 8.8, containing 6 M urea, 30% glycerol, 2% SDS, a trace amount of bromophenol blue, and 45 mg/mL DTT) for 10 min and rinsed in the SDS gel running buffer, followed by running on a 12% SDS PAGE at 15 °C. After the SDS PAGE, the gel images were scanned immediately using the Typhoon TRIO Imager (GE Healthcare). The fluorescent intensities of overlapped spots on the gels were analyzed using DeCyder software. The Phosphor ratio of the two samples (WT/KO) was calculated by adjusting phosphor ratios with protein ratios in LRRK1-deficient vs. control WT cells.

### 2.4. Protein Identification

Differentially phosphorylated spots identified between the two types of cells were collected with an Ettan Spot Picker and digested with Promega Trypsin Gold (Promega, Madison, WI, USA) in the gel. The digested tryptic peptides were desalted with a Zip-tip C18 filter (Millipore, Burlington, MA, USA), eluted from the Zip-tip with 0.5 µL of matrix solution (2-cyano-4-hydroxycinnamic acid 5 mg/mL in 50% acetonitrile, 0.1% trifluoroacetic acid, and 25 mM ammonium bicarbonate) and spotted on the MALDI plate (model ABI 01-192-6-AB).

MALDI-TOF MS and TOF/TOF tandem MS/MS were performed on an AB SCIEX TOF/TOF™ 5800 System (AB SCIEX LLC, Framingham, MA, USA). MALDI-TOF mass spectra were acquired in a reflectron positive ion mode, averaging 4000 laser shots per spectrum. TOF/TOF tandem MS fragmentation spectra were acquired for each sample, averaging 4000 laser shots per fragmentation spectrum on each of the 10 most abundant ions present in each sample (excluding trypsin autolytic peptides and other known background ions). Both the resulting peptide mass and the associated fragmentation spectra were submitted to a GPS Explorer workstation equipped with a MASCOT search engine (Matrix Science, Boston, MA, USA) to search the SwissProt database. Searches were performed without constraining protein molecular weight or isoelectric point, with variable carbamidomethylation of cysteine, oxidation of methionine, and phosphorylation of serine, threonine, and tyrosin, and with one missed cleavage allowed in the search parameters. Candidates with either a protein score of the confidence interval (C.I.) or Ion C.I. greater than 95% were considered significant.

### 2.5. Phosphorylated Protein Validation

An aliquot (30 μg) of the cellular protein was separated on 4–12% SDS NuPAGE (Thermo Fisher Scientific, Waltham, MA, USA), and SuperSep 7.5% Phos-tag (50 µmol/L) SDS-PAGE (Fujifilm Irvine Scientific, Santa Ana, CA 92705, USA), respectively. Phos-tag is a functional molecule that can trap phosphorylated proteins during SDS PAGE and decrease the mobility of phosphorylated proteins, therefore allowing the detection of differentially phosphorylated proteins as a different band. The protein was transferred to a 0.45 µm PVDF membrane, followed by immunoblotting with specific antibodies against candidate proteins for Western blot analysis as described previously [9]. β-Actin was used as a loading control. Intensities of the phosphor-proteins were quantified by Image J software, version 1.54p (National Institutes of Health, Bethesda, MD, USA).

### 2.6. Statistical Analysis

Data were analyzed using a Student’s *t*-test. Data were presented as mean ± standard error of the mean (SEM). Values were considered statistically significant when *p* < 0.05.

## 3. Results

To investigate the mechanisms of LRRK1 regulation of osteoclast function, we performed a proteomic analysis in the differentiated osteoclasts derived from the *Lrrk1* KO and control WT mice (Figure 1A and Appendix A). LRRK1-deficient and WT cells were lysed, and the cellular proteins were labeled with Cy3 (green color) and Cy5 (red color), respectively, 1 h after M-CSF and RANKL re-stimulation. The labeled proteins were mixed, run on the same 2D DIGE, and visualized for protein profiling. The same protein’s red- and green-fluorescent intensities were quantified, and the protein ratio of the WT/KO osteoclasts was calculated. We found no significant change in protein profiling in LRRK1-deficient osteoclasts as compared to WT osteoclasts (Figure 1B). Most protein spots were visible as yellow spots, which resulted from the merging of red and green fluorescence, indicating their equal levels of expression. The protein ratio of WT/KO osteoclasts varied between 0.82 and 1.22 for the selected 71 spots (Figure 1B and Appendix A).

To identify the phosphoproteins regulated by LRRK1 signaling, osteoclasts derived from *Lrrk1* KO or WT mice were lysed, and the same amount of cellular protein was labeled with fluorescent Cy3-dye and separated on a 2D SDS PAGE (Figure 2A). The gels were then stained with Pro-Q^®^ Diamond Phosphoprotein Gel Stain. As shown in Figure 2B, the phosphoproteins were stained as red spots, and the total protein spots were stained green. While the intensity of total proteins (e.g., the green spots in two gels) was comparable, quantitative analysis of the phosphor ratio of each spot between two cell types (WT/KO) uncovered 71 differentially phosphorylated spots that showed a more than 1.5-fold increase (white circles) or decrease (purple circles) (Figure 2B and Appendix A). Among the 71 protein spots, 22 protein spots were shown to be hyperphosphorylated or hypophosphorylated in LRRK1-deficient osteoclasts as compared to WT osteoclasts after adjusting the protein ratio. The intensity of the phosphor-protein ratio of WT/KO was either greater than two-fold or less than 50% in LRRK1-deficient osteoclasts (Table 1 and Appendix A).

To identify which proteins are differentially phosphorylated in LRRK1-deficient osteoclasts, all 22 protein spots were collected and digested with modified porcine trypsin in the gel for mass spectrometer analysis. As shown in Table 1, we identified 21 species of peptides out of 22 spots. Annexin A2 was found in two spots. Sixteen phosphoproteins with phosphorylated peptides were identified. Five proteins were identified without phosphor-sites (e.g., nitric oxide-associated protein 1, transcriptional repressor scratch 2, glutamate-cysteine ligase regulatory subunit, superoxide dismutase, and G protein-regulated inducer of neurite outgrowth 1). While most of the differentially phosphorylated proteins are cellular structural proteins such as tubulins, actins, nucleosome assembly protein 1, and 14-3-3, we found that 6 phosphoproteins (e.g., SNX2, SNX3, VPS35, VTA1, and CTSA and CFL1) which are known to be involved in endosome/lysosome sorting and trafficking were differentially phosphorylated in LRRK1-deficient osteoclasts compared to WT osteoclasts (Table 1). SNX2, VPS35, VTA1, CFL1, and CTSA proteins were significantly hypophosphorylated while SNX3 was hyperphosphorylated in LRRK1-deficient osteoclasts. Phosphorylated VPS35 and CFL1 were reduced by 74% and 60%, respectively, while phosphorylated SNX3 was elevated by four-fold in LRRK1-deficient osteoclasts. Reduced phosphorylation of VPS35 was found at the residues serine 528 and tyrosine 537 while reduced phosphorylation in CFL1 was detected at the residues threonine 148, serine 156, and serine 160 in *Lrrk1* KO osteoclasts. SNX3 was found to be highly phosphorylated at residues serine 79 and serine 85 in LRRK1-deficient osteoclasts (Table 1).

It is known that VPS35, VTA1, SNX2, and SNX3 are critical retromer complex components responsible for membrane trafficking, and CFL1 dynamic cycles of phosphorylation/dephosphorylation are essential for F-actin polymerization and podosome formation. To further validate the protein phosphorylation, we carried out a Western blot analysis. As shown in Figure 3A, expression levels of SNX3, SNX2, pCFL1, VPS35, and VTA1 were comparable between WT and LRRK1-deficient osteoclasts (Figure 3A). There was no change in phosphorylation of CFL1 at residue serine 3 in the *Lrrk1* KO osteoclasts compared with the WT cells, as evidenced by Western blotting with antibody specific to phosphorylated serine 3 of CFL1. To determine if the AKT signal pathway and the Wiskott-Aldrich syndrome protein (WASP), an actin cytoskeleton reorganizer, are altered in *Lrrk1* KO cells, we examined the phosphorylation pattern in AKT, PTEN, GSK-3β, and WASP with specific phosphor antibodies (Figure 3A). Our Western blot showed that there were no changes in levels of pT308-AKT, pS473-AKT, pS9-GSK-3β, pS157-WASP, pS239-WASP, and pS380-PTEN in *Lrrk1* KO osteoclasts compared to WT control cells. There were no changes in the protein expression patterns of CFL1, WASP, and AKT either (Figure 3A).

To further validate phosphorylation in SNX2, SNX3, VPS35, VTA1, and CFL1, we carried out Phos-tag SDS PAGE analyses. Because the Phos-tag acrylamide acts as a phosphate-trapping molecule, the phosphorylated protein migrates slower than non-phosphorylated protein. Concurrent with the findings by phosphor-proteomics analyses, our Phos-tag SDS PAGE analyses also demonstrated alternations in phosphorylation in CFL1 and VPS35 in *Lrrk1* deleted cells. The phosphoprotein levels of CFL1 and VPS35 were reduced by 66% and 69%, respectively, in LRRK1-deficient osteoclasts compared to WT osteoclasts (Figure 3B,C). In contrast with the 2D DIGE phosphor-proteomics analyses, our Pho-tag SDS PAGE analyses failed to detect notable differences in phosphorylated peptides of SNX2, SNX3, and VTA1(Figure 3B).

## 4. Discussion

In this study, we sought to identify phosphoproteins regulated by LRRK1 signaling. Our aim was to explore LRRK1 signal pathways relevant to LRRK1-mediated bone resorptive function in osteoclasts. By 2D DIGE phosphor-proteomics, Phos-tag SDS PAGE, and conventional Western blot analyses, we demonstrated that LRRK1 signaling regulated downstream effectors of VPS35 and CFL1 phosphorylation in osteoclasts. Both VPS35 and CFL1 were highly phosphorylated in the WT osteoclasts but significantly hypophosphorylated in the LRRK1-deficient osteoclasts. Since the dynamic cycles of phosphorylation and dephosphorylation are known to play critical roles in endosome/lysosome trafficking and podosome formation, the difference in the phosphorylation status of VPS35 and CFL1 should affect the lysosomal distribution [12,13]. Our studies show that this is the case. In our previous study, we demonstrated that LRRK1-deficient osteoclasts had an altered position at the acidic vacuoles/lysosomes as well as a weak sealing zone. Lysosomes were distributed away from the ruffled border in LRRK1-deficient osteoclasts [9]. Taken together, our findings indicate that LRRK1 regulates lysosomal sorting, podosome belt formation, and osteoclast bone resorption functions via directly or indirectly modifying VPS35 and CFL1 phosphorylation in osteoclasts.

VPS35 is a critical component of the retromer cargo-selective complex and is implicated in neurodegenerative disease in patients with LRRK2 mutation [14]. The retromer containing a heterotrimeric complex of the subunits VPS35, VPS26, and VPS29 and an associated dimer of the sorting nexins (SNX) transports transmembrane proteins from endosomes back to the trans-Golgi network (TGN), lysosome, or plasma membrane for degradation or recycling [15,16]. Structure–function studies revealed that VPS29 and VPS26 bind to separate regions of VPS35 homodimer, with VPS26 binding to the first 150 residues of the N-terminus and VPS29 binding to the C-terminal residues 517–740 of the VPS35 while the VPS26 contacts both the SNX dimer and VPS35 [17,18,19]. It was reported that phosphorylation of serine 7 of VSP35 by casein kinase (CK2) in adipocytes or muscle cells caused dissociation of the retromer with the microsomal membrane and drives the glucose transporter type 4 (GLUT4) protein trafficking to lysosomes for degradation [20]. Mutation in the CK2 phosphorylation motif of VPS35 (VPS35-S7A) caused VPS35 to be resistant to insulin-induced dissociation from the microsomal membrane, and overexpression of the VPS35-S7A variant attenuated insulin-stimulated GLUT4 degradation [20]. Loss of VPS35 in mice promoted hyper-resorptive osteoclast formation and bone loss via sustained RANKL signaling in osteoclasts [21]. Deficiency of VPS35 in osteoblasts impaired parathyroid hormone (PTH)-promoted PTH type 1 receptor trafficking to TGN, enhanced PTH signaling [22]. These studies indicate that VPS35 is a negative regulator of membrane receptors, and phosphorylation of VPS35 promotes the translocation of the receptors to lysosomes for degradation.

In our previous studies, we found that LRRK1 depletion in mice provokes osteoclast dysfunction due to the failure of osteoclasts to distribute lysosomes to ruffled borders in response to RANKL stimulation [9]. We assumed that a lack of LRRK1 may downregulate the phosphorylation of proteins critical for lysosomal sorting. As expected, our study revealed that phosphorylation of VPS35 at residue serine 528 in the region to which VPS29 binds was remarkably attenuated in LRRK1-deficient osteoclasts. The physiological role of the serine 528 dephosphorylation of VPS35 in osteoclasts is currently unknown. It is possible that phosphorylation of VPS35 at serine 528 is required for VPS35 dimerization or interaction of VPS35 with VPS29. Alternatively, the phosphorylation of serine 528 is necessary to dissociate the retromer with the microsomal membrane and allow the membrane receptor to sort to lysosomes for degradation. Dephosphorylation of VPS35 in LRRK1-deficient cells is assumed to impair the retromer’s function. Based on the findings that VPS35 was a negative regulator of RANKL signaling and that dephosphorylation of VPS35 decreased transportation of membrane receptors to lysosomes for degradation, we should expect a sustained RANKL-induced osteoclast activity in LRRK1-deficient osteoclasts. As anticipated, we did observe the giant osteoclasts in the long bone of *Lrrk1 KO* mice and in vitro osteoclast cultures. However, in contrast to our expectation these LRRK1-deficient osteoclasts were dysfunctional [8]. Our study indicates that VPS35 dephosphorylation only enhances RANKL-induced osteoclast differentiation but does not accelerate osteoclast function in the absence of LRRK1. Another LRRK1 targeted protein may play a crucial role in regulating osteoclast function and bone resorption.

The non-muscle type CFL1 is an actin-binding protein involved in dynamic remodeling of the actin cytoskeleton [23]. Active CFL1 promotes actin filament disassembly by binding to ADP-bound F-actin, severing the filament, and releasing actin monomers, thereby accelerating actin cytoskeleton dynamics [13,24,25]. Phosphorylation at serine 3 of CFL1 was reported not to bind to either F-actin or G-actin, and as such stabilizes the polymerized actin cytoskeleton while unphosphorylated CFL binds to both and disrupts actin structure and causes the formation of cytoplasmic actin bundles in 293 cells [25]. In this study, we did not detect a significant difference in N-terminal serine 3 phosphorylation, but we uncovered a remarkable reduction in the C-terminal phosphorylation at the residues threonine 148, serine 156, and serine 160 of CFL1 in LRRK1-deficient osteoclasts. Because conditional disruption of *Cfl1* gene in osteoclasts in mice resulted in osteoclast dysfunction and reduced bone resorption via compromising podosome patterning [26], it is reasonable to assume that C-terminal dephosphorylation might, like the N-terminus, also contribute to impaired podosome/sealing zone formation and bone resorption in the LRRK1-deficient osteoclasts. Consistent with this prediction, we observed impaired F-actin ring formation and sealing zones in *Lrrk1* KO mice in our previous report [8]. However, the role of CFL1 C-terminal phosphorylation in regulating cytoskeleton rearrangement is elusive. Thus, further studies are needed to determine if the LRRK1 regulation of CFL1 C-terminal phosphorylation is a mechanism for the cytoskeleton rearrangement in osteoclasts.

The signal pathway underlying LRRK1-mediated phosphorylation of CFL1 in osteoclasts is currently undefined. It has been known that several protein kinases and phosphatases regulate the phosphorylation level of CFL1 at the residue serine 3 in various types of cells. The major kinases of CFL1 are LIM-kinases (LIMKs), testicular protein kinases (TESKs), and Nck-interacting kinase-related kinase (NRK) [27,28,29]. Besides the serine 3 residue, CFL1 was also phosphorylated at threonine 63, tyrosine 82, and serine 108 in myeloid cells treated with LIM kinase inhibitor [30]. In addition, ERK1/2 kinases have been reported to phosphorylate threonine 25 in CFL1 in heart tissue, leading to alterations in left ventricular function and cardiac actin [31]. Interestingly, LIMKs are phosphorylated and activated by p21-activated kinase 1 (PAK1), which is a downstream effector of AKT/small GTPase proteins, RAC1 and Cdc42, signal axis [32,33]. On the other hand, CFL1 is dephosphorylated by members of the Slingshot (SSH) protein phosphatase family, protein phosphatase 1 (PP1), and PP2A [34,35]. The phosphorylation of SSH1 and PP1 at serine or threonine residues could inactivate the phosphatase activity [36,37]. In the present study, we examined AKT/GSK3β and found no changes in the activation of AKT and GSK3β in the LRRK1-deficient osteoclasts. We did not detect a change in PTEN phosphorylation either. How the C-terminal CFL1 obtains phosphorylated is unknown. We assumed that LRRK1 may regulate the CFL1 serine/threonine phosphorylation by modulating RAC1/PAK1/LIMKs signals since our previous study demonstrated that LRRK1 regulated osteoclast function by modulating RAC1/CDC42 small GTPase phosphorylation and downstream PAK1 activation which is upstream of the effectors of LIMKs [28,38,39]. The phosphorylation levels of RAC1/CDC42 and PAK1 were remarkedly compromised in LRRK1-deficient osteoclasts [38]. However, it is also possible that LRRK1 modulates CLF1 phosphorylation by phosphorylating SSHs, PP1, or PP2A, as such inactivating phosphatase activity [35,36,37]. Alternatively, LRRK1 may directly phosphorylate CFL1 in osteoclasts. Further studies are needed to explore these possibilities.

While we detected other phosphorylated proteins of SNX2, SNX3, and VTA1 by proteomics analysis, we failed to validate the phosphorylation by Phos-tag SDS analyses. The potential reasons for this discrepancy include the following: First, MALDI-TOF MS and TOF/TOF tandem MS/MS analyses were sensitive enough to detect trace amounts of phosphorylated peptides with low specificity, but Phos-tag SDS PAGE analysis was not. Second, some pan antibodies may not be able to recognize the antigens of phosphorylated proteins because of a conformation change. Whether the SNX2, SNX3, and VTA1 are phosphorylated in osteoclasts and involved in the regulation of lysosome trafficking and function needs to be studied in the future.

There are a few limitations in this study. First, the physiological roles of the C-terminal phosphorylation of VPS35 and CFL1 in osteoclasts in regulating osteoclast function are not validated. Second, we did not determine if the C-terminal phosphorylation of CFL1 and VPS35 is osteoclast-specific and differentiation stage dependent. Third, it remains undetermined if VPS35 and CFL1 are direct biological substrates of LRRK1 in osteoclasts. In our future studies, we will make a phosphomimic of VPS35-S528D and a dephosphomimic of VPS35-S528A and overexpress them in LRRK1-deficient and WT osteoclasts to determine the mutant variants’ function. Similarly, we will perform site-directed mutagenesis for CFL1 expression vectors and evaluate the phosphomimic and dephosphomimic CFL1’s function. We will also examine differences in LIMKs, SHHs, and PP1 phosphorylation in LRRK1-deficient osteoclasts, search for direct biological substrates in osteoclasts, and carry out kinase assays to determine if purified recombinant protein LRRK1 can phosphorylate LIMKs, SSHs, PP1, CFL1, and VPS35.

## 5. Conclusions

In this study, we performed 2D DIGE phosphor-proteomics, mass spectrometry, and Phos-tag SDS PAGE analysis to identify LRRK1 targets in osteoclasts. We found that C-terminal phosphorylation of CFL1 and VPS35 was significantly compromised in LRRK1-deficient osteoclasts. Dephosphorylation at serine 528 of the VPS35 is predicted to cause sustained RANKL signaling and increase osteoclast formation while dephosphorylation at threonine 148, serine 156, and serine 160 of CFFL1 may lead to a disorganized cytoskeleton and osteoclast dysfunction. Our study, together with others, supports our predicted mechanisms of LRRK1 action in osteoclasts (Figure 4A,B). In the WT osteoclasts, LRRK1 directly or indirectly phosphorylates VPS35 and CFL1 proteins, leading to late endosome/lysosome trafficking to the ruffled border, F-actin polymerization/depolymerization cycling, podosome formation, and bone resorption. In the absence of LRRK1 signaling, osteoclasts cannot form functional podosomes but have sustained RANKL signaling to continue to fuse, making the dysfunctional osteoclast larger as shown in Figure 4A. Further studies are needed to confirm if the VPS35 and CFL1 as well as the upstream kinases or phosphatases are direct biological substrates of LRRK1 in osteoclasts.

## Figures and Tables

**Figure 1 biology-14-00326-f001:**
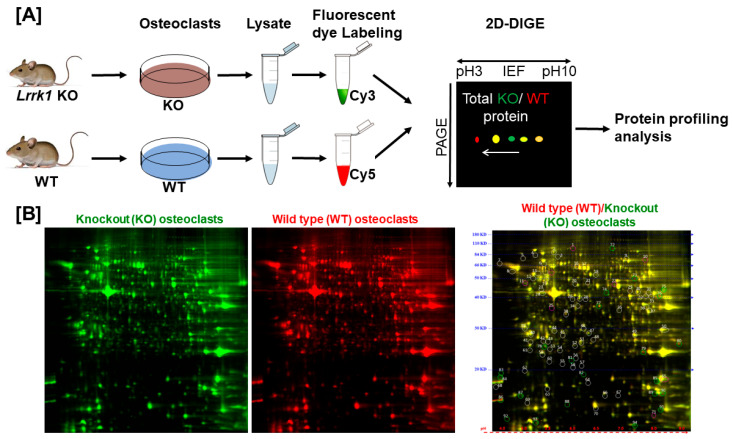
There was no significant change in protein profiling analysis in LRRK1-deficient osteoclasts. (**A**) A schematic procedure of the experiments for protein profiling analyses. (**B**) Protein profiling of the LRRK1-deficient (KO) and wild-type (WT) osteoclasts as well as the overlapping of KO and WT cells. The cellular proteins derived from *Lrrk1* KO and WT osteoclasts were labeled with Cy3 (green color) and Cy5 (red color), respectively.

**Figure 2 biology-14-00326-f002:**
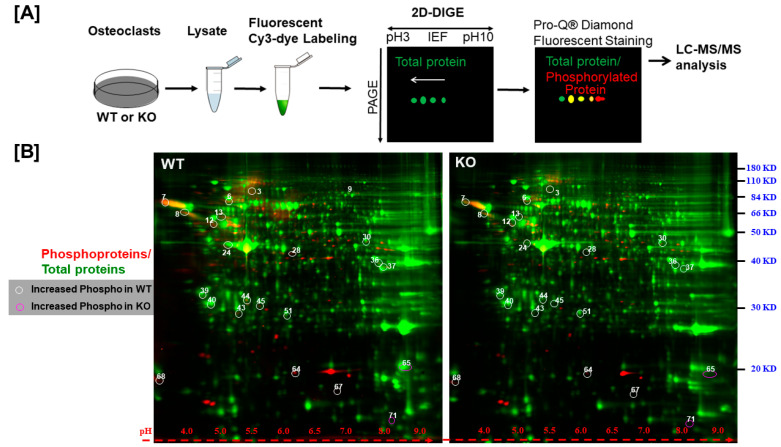
Proteins are differentially phosphorylated in LRRK1-deficient osteoclasts. (**A**) A schematic procedure of the experiments for phosphor-proteomics analyses and candidate protein identification. (**B**) Images of total proteins/phosphor-proteins on 2D gel. The 71 differentially phosphorylated proteins were numbered in the gel. Total cellular proteins were labeled with green color, and the phosphoproteins were stained with red color.

**Figure 3 biology-14-00326-f003:**
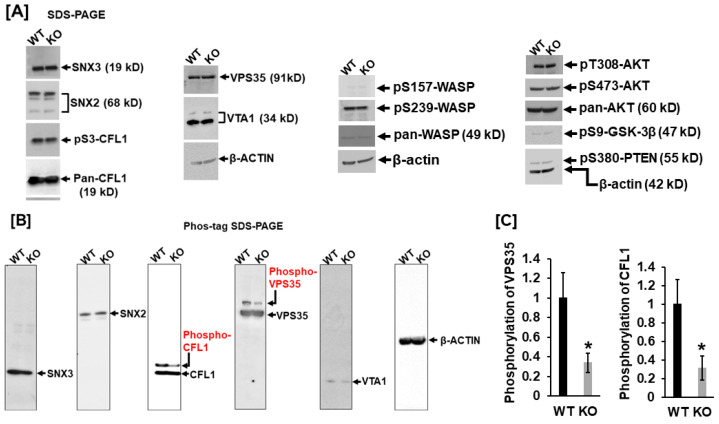
Phos-tag SDS-PAGE analyses confirm hypophosphorylation of cofilin 1 (CFL1) and vacuolar protein sorting-associated protein 35 (VPS35) in LRRK1-deficient osteoclasts. (**A**) Immunoblotting analyses with antibodies specific to phosphorylated peptides. (**B**) Immunoblotting analyses after Phos-tag SDS-PAGE. The retarded bands of phosphorylated CFL1 and VPS35 proteins are indicated in red. (**C**) Quantification of relative phosphorylation of VSP35 and CFL1, respectively, in WT and LRRK1-deficient osteoclasts (*n* = 3), and a star indicates *p* < 0.05.

**Figure 4 biology-14-00326-f004:**
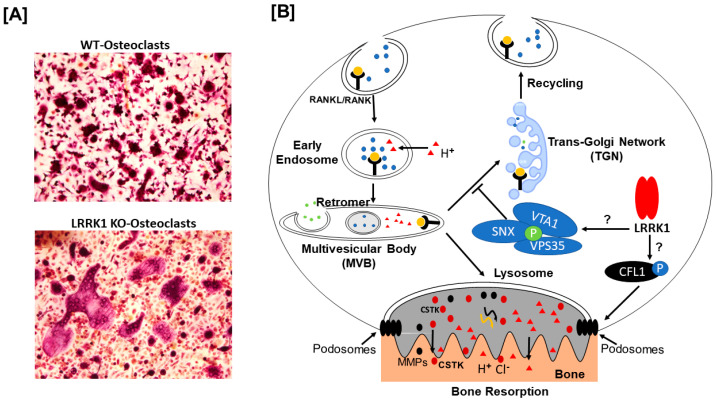
Proposed mechanisms of LRRK1 action in osteoclasts. (**A**) Representative TRAP-positive osteoclast cultures on bone slices. Osteoclasts derived from LRRK1-deficient mice are flat and larger than the WT osteoclasts on the bone slice. (**B**) A schematic diagram of LRRK1 action in osteoclasts. LRRK1 directly or indirectly phosphorylates VPS35 and CFL1 proteins in osteoclasts, leading to late endosome/lysosome trafficking to the ruffled border, F-action polymerization, podosome formation, and bone resorption. Lack of LRRK1 signaling causes endosome/lysosome trafficking to the trans-Golgi network (TGN), RANKL signaling termination, and membrane protein recycling, leading to continuing fusion of osteoclasts without bone resorption.

**Table 1 biology-14-00326-t001:** Identification of phosphorylated proteins in WT and LRRK1-deficient osteoclasts.

Spot ID	Protein Ratio WT/KO	Phospho-Ratio WT/KO	Protein Name	Identified Peptide	Phospho-Site
3	0.96	3.70	Vacuolar protein sorting-associated protein 35	IRFTLPPLVFAAYQLAFR	T528, Y537
6	0.96	2.70	Sorting nexin-2	SIVGMTK, AALERYLQR	S212, Y242
				VGKEDSSSTEFVEK, MNESDAWFEEK	S226, S227, S302
8	1.11	4.44	Nucleosome assembly protein 1	QVPNESFFNFFSPLK	S288, S294
12	0.93	3.75	Tubulin beta-5 chain	YVPRAILVDLEPGTMDSV,	Y59, T72
				IREEYPDR, LHFFMPGFAPLTSR	Y159, T274, S275
13	0.97	3.42	Tubulin alpha-1C chain	NLDIERPTYTNLNR, DVNAAIATIK	T223, Y224, T225, T334
22	1.03	0.40	Alpha-enolase	SILRIHAR, FGANAILGVSLAVCK	S2, S115
24	1.02	4.69	Actin, cytoplasmic 2	DSYVGDEAQSKR, GYSFTTTAEREIVR,	S51, Y52, Y198, S199
				KDLYANTVLSGGTTMYP, EITALAPSTMKIK	Y294, T297, S300, Y306, T318
28	0.89	2.33	Vacuolar protein sorting-associated protein VTA1	YARWK, AQKYCK	Y144, Y277
30	1.06	2.13	Isocitrate dehydrogenase [NADP]	GQETSTNPIASIFAWSR, SDYLNTFEFMDK	T325, S326, T327, S389, Y391, T394
36	0.95	2.36	Annexin A2	KELPSALK, KGTDVPK, LYDSMK, SEVDMLK	S85, T208,Y275, S277, S296
37	0.98	3.15	Annexin A2	ELYDAGVKR, WISIMTERSVCHLQK	Y199, S215,T218
39	0.89	4.19	Lysosomal protective protein (Cathepsin A)	FPEALMRSGDK, ECSHITFLTIK	S315, S439, T442, T445
40	0.88	2.39	14-3-3 protein eta	MKGDYYR, YLAEVASGEKK	Y130, Y131, Y133
45	1.02	2.59	Actin, cytoplasmic 1	KDLYANTVLSGGTTMYP, MQKEITALAPSTMK	Y294, T297, S300, Y306, T318, T324
51	0.93	2.46	Cathepsin S	GCVTEVK, ATDEKCHYNSK, NNKNHCGIASYCSYPEI	T136, T218, Y224 S333, Y334, S336
64	0.93	2.45	Cofilin 1	CTLAEKLGGSAVISLEGK	T148, S156, S160
65	0.83	0.25	Sorting nexin-3 (could be Cofilin-1)	SELERESK (SNX3)	S79, S85

Red color characters are detected phosphorylated residues.

## Data Availability

Data presented in this study are availability from PI upon request.

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
