# Peer review of "Leucine-Rich Repeat Kinase 1 Signaling Targets Proteins Critical for Endosome/Lysosome Sorting and Trafficking in Osteoclasts†"

_biology, 2025, doi:10.3390/biology14040326_

Round 1

Reviewer 1 Report

Comments and Suggestions for Authors

The paper of Xing et al. is interesting and could be considered for publication after  major revisions. Several experiments are needed to better support the proposed mechanism of action of LRRK1 in osteoclasts and to clarify some questions raised. Many sentences in the discussion are not sufficiently supported by experimental evidence. Additionally, there is a lack of statistical analysis, the blots are not well presented, and the number of replicates is not mentioned. These issues should be resolved.

Introduction:

Line66: to

Line 70-73: where are the bibliographic citations?

In the introduction, the authors mention two diseases related to osteoclasts; however, it is not entirely clear what are the specific differences between these pathologies or how they contrast in terms of molecular mechanisms. Additionally, the role of LRRK1 in osteoporosis is not explicitly defined, leading to some confusion about its relevance in the osteoclasts. It is not evident why authors start the introduction with osteoporosis and conclude with osteopetrosis. A connection between these two diseases and how they support the central hypothesis of the study would be beneficial for the reader.

Methods:

Line110-111: “The cells were than starved in M-CSF and RANKL-free media for 1 hour, followed by 1 hour M-CSF 110 and RANKL stimulation.” Why was the starvation period limited to only one hour, and what were the concentrations of M-CSF and RANKL used for stimulation?

Moreover, authors did not specify the number of mice per group or the total number of groups included in the study. Was the incubation time sufficient to obtain mature osteoclasts, and how was this maturity confirmed?

Line 121-122: The sentence is not clear. Was each group mixed with 1 μL of Cy3 ?

Line 129: respectively. “The Labeled proteins were mixed together”. What authors mean by labeled proteins were mixed together? Did they load samples onto different gels? This is what we inferred from the results section.

Results:

Line 173: Cy3 (shown in red) and Cy5 (shown in green), Please review this line, it does not align with the schematic procedure.

Please review table 1 and correct some errors found.

It would be interesting to determine whether this difference in the phosphorylation of these proteins affects the distribution and morphology of endosomes and lysosomes in differentiated osteoclasts.

What is the difference between male and female samples?

It would be important to clarify whether authors detected SNX3 (19 kDa) and VPS35 on the same membrane, as this could impact the interpretation of the results. Additionally, no specific loading control for SNX3 is observed, which is essential to ensure the reliability and proper normalization of the presented data.

In figure B, it is unclear why beta-actin, used as a loading control, is presented alone. Ideally, a specific loading control should be provided for each detected protein if they are from different membranes, along with the corresponding densitometric analysis and statistical data to validate the reported differences. Additionally, results should include an adequate number of replicates (at least n=3) to ensure the statistical analysis of the findings.

Discussion:

Lines 253-256: It would be interesting to study the distribution and morphology of endo-lysosomal organelles in LRRK1-deficient osteoclasts to support well the discussion.

Additionally, I recommend the authors to analyze the downstream signaling pathways of RANKL in these LRRK1-deficient cells, at least AKT or MAPK should be addressed to elucidate potential signaling defects.

More experiments using the cell model are required to establish the role of LRRK1 in osteoclasts. Authors should determine the levels of cathepsin( one of them), the colocalization of relevant membrane proteins with LAMP2.

Finally, please review and correct some words and support the data by statistical analyses.

Author Response

We thank the reviewer 1 for his/her positive remarks.

  1. Line66: to

Response: We apologize for the error. We have corrected the error in the revised manuscript.

  1. Line 70-73: where are the bibliographic citations?

Response: We apologize for missing the citation. We have now added reference in the revised manuscript.

  1. In the introduction, the authors mention two diseases related to osteoclasts; however, it is not entirely clear what are the specific differences between these pathologies or how they contrast in terms of molecular mechanisms. Additionally, the role of LRRK1 in osteoporosis is not explicitly defined, leading to some confusion about its relevance in the osteoclasts. It is not evident why authors start the introduction with osteoporosis and conclude with osteopetrosis. A connection between these two diseases and how they support the central hypothesis of the study would be beneficial for the reader.

Response: We thank the reviewer for his/her suggestion. We have now made the osteoporosis vs. osteopetrosis clear in the revised manuscript. Osteoporosis is a bone disease with bone loss or low bone density while osteopetrosis is a bone disease that causes bone to grow abnormally and overly dense. Mice without LRRK1 causes high bone density. Thus, it is an ideal drug target. Partially inhibition of LRRK1 function with small molecules is assumed to enhance the bone density for osteoporosis treatment.

  1. Line110-111: “The cells were than starved in M-CSF and RANKL-free media for 1 hour, followed by 1 hour M-CSF 110 and RANKL stimulation.” Why was the starvation period limited to only one hour, and what were the concentrations of M-CSF and RANKL used for stimulation?

Response: We apologize for the missing information and an error. We have now added the information in the revised manuscript. We only starved the cell for 1 hours to prevent mature osteoclasts from apoptosis. Final concentration of M-CSF is 20 ng/ml and final concentration of RANK-GST fusion protein is 200 ng/ml.

  1. Moreover, authors did not specify the number of mice per group or the total number of groups included in the study. Was the incubation time sufficient to obtain mature osteoclasts, and how was this maturity confirmed?

Response: We apologize for the missing information. We collected and cultured osteoclasts derived from 3 KO females and 3 males and corresponding WT female and male littermates. However, we combined the lysates of osteoclasts from the same animal group for proteomics analyses. We differentiate the splenocytes in the presence of M-CSF and RANKL for 5 days as described in the materials and methods. Images of differentiated and undifferentiated cells stained with TRAP, an osteoclast marker, are provided in the revised manuscript.

  1. Line 121-122: The sentence is not clear. Was each group mixed with 1 μL of Cy3 ?

Response: We apologize for not making the sentence clear. We have now revised and made it clear in the revised manuscript. Protein from Lrrk1 KO and WT osteoclasts was mixed with 1.0 µl of diluted Cy3-dye (fluorescents greenish yellow), respectively.

  1. Line 129: respectively. “The Labeled proteins were mixed together”. What authors mean by labeled proteins were mixed together? Did they load samples onto different gels? This is what we inferred from the results section.

Response: W apologize for unclear. We have not revised the sentence in the revised manuscript. We stated that “For the protein profiling, 30 µg cellular protein from Lrrk1 KO and control WT cells was labelled with Cy3- and Cy5-dye, respectively. The Cy3-labelled Lrrk1 KO and Cy5-Labelled WT proteins were mixed. The mixture of two labelled proteins was loaded into the same IEF strip holder and run on the same gel (pH3-10 linear) according to Amersham BioSciences’ instructions.”

  1. Line 173: Cy3 (shown in red) and Cy5 (shown in green), Please review this line, it does not align with the schematic procedure.

Response: We apologize for the mistake. We have now corrected the error in the revised manuscript.

  1. Please review table 1 and correct some errors found.

Response: We apologize for the errors of the phosphor-site in the table. We have now corrected the errors and organized them in order in the revised Table 1.

  1. It would be interesting to determine whether this difference in the phosphorylation of these proteins affects the distribution and morphology of endosomes and lysosomes in differentiated osteoclasts.

Response: We agree with the reviewer that difference in the phosphorylation of proteins should affect the endosomes/lysosomes distribution in osteoclasts. Consistent with this prediction, our previous studies have demonstrated that LRRK1 deficient osteoclast, unlike the wild-type (WT) osteoclast, had an altered trafficking of the acidic vacuoles/lysosomes (8). The lysosomes were distributed in the cytoplasmic compartment away from the extracellular lacunae of the bone. Although the expression of cathepsin K and v-ATPase were comparable in LRRK1 deficient osteoclasts, these proteins were not localized on the ruffled border and were not secreted. We have included this information in the introduction with reference 8. Further studies are needed to test if the over expression of phosphor-mimic proteins in Lrrk1 knockout cells can rescue the bone resorption function or not in an in invitro pit formation assay. We added this information in the discussion in the revised manuscript.

  1. What is the difference between male and female samples?

Response: We did not see any difference in in vitro differentiation or in vitro bone resorption function of osteoclasts derived from male and female mice. Therefore, we combined cells from 3 females and 3 males to get enough cellular protein for a series of proteomics analyses.

  1. It would be important to clarify whether authors detected SNX3 (19 kDa) and VPS35 on the same membrane, as this could impact the interpretation of the results. Additionally, no specific loading control for SNX3 is observed, which is essential to ensure the reliability and proper normalization of the presented data.

Response: They were from the same gel and the same membrane with the same amount protein. We prepared 4 strips, each containing same protein from WT and KO. Strips were blotted 2-3 times with different antibodies if the sizes of interest proteins are different as shown and described in the original blots. While we did re-blot all membrane trips with beta-actin, the intensities of the bands were comparable among the strips. Therefore, only one load control was presented in the figure.

  1. In figure B, it is unclear why beta-actin, used as a loading control, is presented alone. Ideally, a specific loading control should be provided for each detected protein if they are from different membranes, along with the corresponding densitometric analysis and statistical data to validate the reported differences. Additionally, results should include an adequate number of replicates (at least n=3) to ensure the statistical analysis of the findings.

Response: We agree with the reviewer. Please see response 12. Intensities of the signals are now provided with replicates of CFL1 and VPS35 that are shown differences in phosphorylation in Figure 4 in the revised manuscript.

  1. Lines 253-256: It would be interesting to study the distribution and morphology of endo-lysosomal organelles in LRRK1-deficient osteoclasts to support well the discussion.

Response: We agree with the reviewer. However, this part of work has been published (reference 8). Please see response 10.

  1. Additionally, I recommend the authors to analyze the downstream signaling pathways of RANKL in these LRRK1-deficient cells, at least AKT or MAPK should be addressed to elucidate potential signaling defects.

Response: We agree with the reviewer. We have analyzed various signal pathways that might regulate cytoskeletal arrangement including AKT, GSK, PTEN, and VASP. We have now provided these data in Figure 4 in the revised manuscript.

  1. More experiments using the cell model are required to establish the role of LRRK1 in osteoclasts. Authors should determine the levels of cathepsin (one of them), the colocalization of relevant membrane proteins with LAMP2.

Response: We agree with the reviewer. We have done the work of cathepsin K trafficking and published in the data in our previous publication (reference 8). Please also see the response 10.

  1. Finally, please review and correct some words and support the data by statistical analyses.

Response: We apologize for the errors, typos, and missing statistical analysis. We have now corrected the errors and typos. We have also quantified the data and performed statistical analyses.

Reviewer 2 Report

Comments and Suggestions for Authors

In this short article, the authors try to identify potential LRRK1 targets in osteoclasts by performing 2D phosphor proteomics analysis with lysates extracted from osteoclasts derived from LRRK1 knockout and wild type mice. Although the proteomics part was reasonably well executed, the overall conclusions are presently hard to draw since no follow-up experiments were performed.

I have the following critique for the manuscript:

1.     Please include some data including images of cells pre-and post- differentiation in the presence of M-CSF and RANKL for both LRRK1 knockout and wild type mice. Either use markers of differentiation and images for osteoclasts or other measures as shown in this paper: https://pubmed.ncbi.nlm.nih.gov/21738673/

2.     There is a typo in line 66, where it says ‘there is a critical need to for development’. It should be: ‘There is a critical need for the development’.

3.     There's another typo in line 90 where it says RANKL-GST fustion instead of ‘GST fusion’.

4.     The method section needs to be more detailed. Specifically, section 2.1, where dilutions for primary antibodies have not been specified. For each antibody, please do so for other labs to be able to replicate your experiments. Also specify the catalog number and name for the HRP-conjugated secondary antibodies used for all the primary antibodies.

5.     There is another typo in line 99, where ‘manufacturers’ is misspelled.

6.     In the results section, line 173 says Cy3(shown in red) and Cy5(shown in green). I believe it should be the opposite.

7.     In line 236, 'Phos- tag SDS-PAGE' is misspelled.

8.     Figure 3 has been mislabeled as Figure 4, and Figure 4 has been mislabeled as Figure 5.

9.     In lines 236 to 238, you mention not being able to capture the phosphorylation of CFL1 with specific antibodies, but the figure shows a phosphorylation band for that protein. Please clarify.

10.  I liked your suggested experiment in the discussion section. Can you include this in vitro kinase assay to further substantiate the conclusions drawn about CFL1 and VPS35 to be included in this paper.

11.  In line 345, there is no space between LRRK1 and deficient.

12.  In line 348, 'threonine' is misspelled.

Author Response

  1. Please include some data including images of cells pre-and post- differentiation in the presence of M-CSF and RANKL for both LRRK1 knockout and wild type mice. Either use markers of differentiation and images for osteoclasts or other measures as shown in this paper: https://pubmed.ncbi.nlm.nih.gov/21738673/

Response: We thank the reviewer for her/his comments. We have now provided these images of osteoclast precursors cultured in the absence and presence of RANKL and M-CSF for 5 days in the revised manuscript. The cells are stained with osteoclast differentiation marker, TRAP.  In our previous publications, we have provided the in vitro differentiation and functional pit formation of osteoclasts derived from Lrrk1 KO and WT control mice both in the plastic dishes and on the bone slices.

  1. There is a typo in line 66, where it says ‘there is a critical need to for development’. It should be: ‘There is a critical need for the development’.

Response: We apologize for the error. We have now corrected the error in the revised manuscript.

  1. There's another typo in line 90 where it says RANKL-GST fustion instead of ‘GST fusion’.

Response: We apologize for the typo. We have now corrected the typo in the revised manuscript.

  1. The method section needs to be more detailed. Specifically, section 2.1, where dilutions for primary antibodies have not been specified. For each antibody, please do so for other labs to be able to replicate your experiments. Also specify the catalog number and name for the HRP-conjugated secondary antibodies used for all the primary antibodies.

Response: We apologize for the missing antibody sources and dilution we applied. We have now provided this information in the revised manuscript.

  1. There is another typo in line 99, where ‘manufacturers’ is misspelled.

Response: We apologize for the misspelling. We have now corrected the error.

  1. In the results section, line 173 says Cy3(shown in red) and Cy5(shown in green). I believe it should be the opposite.

Response: We thank the reviewer for pointing out the errors. We have now corrected the errors in the revised manuscript.

  1. In line 236, 'Phos- tag SDS-PAGE' is misspelled.

Response: We have now corrected the error in the revised manuscript.

  1. Figure 3 has been mislabeled as Figure 4, and Figure 4 has been mislabeled as Figure 5.

Response: We have now corrected the mislabeling in the revised manuscript.

  1. In lines 236 to 238, you mention not being able to capture the phosphorylation of CFL1 with specific antibodies, but the figure shows a phosphorylation band for that protein. Please clarify.

Response: We apologize for not clarification. We have now clarified the sentence by stating that there was no difference in phosphorylation of CFL1 at residue of serine 3 between the WT and LRRK1 deficient osteoclasts.

  1. I liked your suggested experiment in the discussion section. Can you include this in vitro kinase assay to further substantiate the conclusions drawn about CFL1 and VPS35 to be included in this paper.

Response: We thank the reviewer for her/his suggestion. We have not included the in vitro kinase assay to determine if CFL1 and/or VPS35 are direct biological substances of LRRK1 in the discussion in the revised manuscript.

  1. In line 345, there is no space between LRRK1 and deficient.

Response: We have now corrected in the error in the revised manuscript.

  1. In line 348, 'threonine' is misspelled.

Response: We have corrected the misspelling in the revised manuscript.

Round 2

Reviewer 1 Report

Comments and Suggestions for Authors

The english should be carefully revised before publication.

F-actin, etc...

Comments on the Quality of English Language

The english should be revised before publication.

Author Response

The english should be carefully revised before publication. F-actin, etc...

Response: We apologize for errors and typos. We have now corrected these errors and typos in the revised manuscript.

Reviewer 2 Report

Comments and Suggestions for Authors

·      In line 80, ‘resisted to’ looks like a typo.

·      There are several typos in Methods section 2.2 in lines 114, 118 and 120.

·      Line 133, it should be ‘protein concentration’ and not ‘Protein concentration’

·      Line 187, it is ‘Student’s t-test’ and not ‘Students t test’

·      I also believe the kinase assay mentioned in the discussion section is important to come up with the hypothesis of how mechanisms of LRRK1 functions in osteoclasts. Without this assay, this research becomes a proteomics paper where mechanisms of action were not explored. Please consider running that or a similar in-vitro assay to explore the function of LRRK1

Author Response

  • In line 80, ‘resisted to’ looks like a typo. There are several typos in Methods section in lines 114, 118 and 120. Line 133, it should be ‘protein concentration’ and not ‘Protein concentration. Line 187, it is ‘Student’s t-test’ and not ‘Students t test’.

Response: We apologize for the errors and typos. We have now corrected these errors and typos in the revised manuscript.

  • I also believe the kinase assay mentioned in the discussion section is important to come up with the hypothesis of how mechanisms of LRRK1 functions in osteoclasts. Without this assay, this research becomes a proteomics paper where mechanisms of action were not explored. Please consider running that or a similar in-vitro assay to explore the function of LRRK1.

Response: We agree with the reviewer’s suggestion that it is better to explore the LRRK1 signal pathways in osteoclast. LRRK1 could phosphorylate the CFL1 and/or VPS35 in osteoclasts. It is also possible to directly target protein kinases or protein phosphatases that modify CFL1 or VPS35. We have included the possibility in the discussion in the revised manuscript. We stated: “The LRRK1-mediated phosphorylation of cofilin in osteoclasts are currently undefined yet. Previous studies have demonstrated that the phosphorylation level of CFL1 at residue of serine 3 is regulated by several protein kinases and phosphatases in various types of cells. The major kinases of CFL1 are LIM-kinases (LIMKs), testicular protein kinases (TESKs), and Nck-interacting kinase related kinase (NRK) [27-29]. Besides serine 3 phosphorylation, CFL1 was also found to be phosphorylated at other residues of threonine 63, tyrosine 82, and serine 108 in myeloid cells treated with LIM kinase inhibitor [30]. In addition, ERK1/2 kinases have been reported to phosphorylate threonine 25 in CFL1 in hear tissue, leading to alterations in left ventricular function and cardiac actin [31]. Interestingly, LIMKs are phosphorylated and activated by p21 activated kinase 1 (PAK1), which is downstream effector of AKT/small GTPase proteins, RAC1 and Cdc42, signal axis [32, 33]. On the other hand, CFL1 is dephosphorylated by members of the Slingshot (SSH) protein phosphatase family, protein phosphatase 1 (PP1), and PP2A [34, 35]. In the present study, we examined the AKT/GSK3β and found no changes in activation of AKT and GSK3β in the LRRK1 deficient osteoclasts. However, it is possible that LRRK1 regulates CFL1 C-terminal serine/threonine phosphorylation by modulating RAC1/LIMKs signals or phosphatases activity or via directly phosphorylating CFL1 in osteoclasts. Further studies are needed to determine the possibility.”